# The Potential Benefits of Vonoprazan as *Helicobacter pylori* Infection Therapy

**DOI:** 10.3390/ph13100276

**Published:** 2020-09-28

**Authors:** Muhammad Miftahussurur, Boby Pratama Putra, Yoshio Yamaoka

**Affiliations:** 1Division of Gastroenterology and Hepatology, Department of Internal Medicine, Faculty of Medicine, Universitas Airlangga, Dr. Soetomo General Hospital, Surabaya 60286, Indonesia; 2Institute of Tropical Disease, Universitas Airlangga, Surabaya 60132, Indonesia; 3Faculty of Medicine, Universitas Airlangga, Surabaya 60115, Indonesia; boby.pratama-13@fk.unair.ac.id; 4Department of Environmental and Preventive Medicine, Oita University Faculty of Medicine, Yufu 879-5593, Japan; yyamaoka@oita-u.ac.jp

**Keywords:** *Helicobacter pylori*, acid suppression agents, proton pump inhibitor, potassium-competitive acid blocker, vonoprazan

## Abstract

*Helicobacter pylori* infection is a severe global health problem that is closely associated with acid-related diseases and gastric malignancies. Eradicating *H. pylori* is strongly recommended for lowering peptic ulcer recurrence and preventing gastric cancer. The current approved *H. pylori* eradication regimen combines a proton pump inhibitor (PPI) with two antibiotics. Unfortunately, this regimen failed to meet expectations mostly due to antibiotic resistance and insufficient gastric acid suppression. Vonoprazan, a novel potassium-competitive acid blocker, showed promising results as a PPI replacement. Vonoprazan inhibits gastric acid secretion by acting as a reversible competitive inhibitor against potassium ions and forming disulfide bonds with the cysteine molecule of H^+^/K^+^-ATPase. Vonoprazan has superior pharmacological characteristics over PPI, such as no requirement for acid activation, stability in acidic conditions, shorter optimum acid suppression period, and resistance to cytochrome P (CYP)2C19 polymorphism. Several comparative randomized controlled trials and meta-analyses revealed the superiority of vonoprazan in eradicating *H. pylori*, notably the resistant strains. The adverse effect caused by vonoprazan is long-term acid suppression that may induce elevated gastrin serum, hypochlorhydria, and malabsorption. All vonoprazan studies have only been conducted in Japan. Further studies outside Japan are necessary for universally conclusive results.

## 1. Introduction

*Helicobacter pylori* is a unique, human-specific pathogen that can be found in the human stomach in about 40–50% of the global population. *H. pylori* infection is a significant global health problem whose prevalence is about 44.3%, from 34.7% in developed countries to 50.8% in developing countries, with a global recurrence rate of 4.3–4.6% [1,2,3]. An epidemiologic meta-analysis study revealed that *H. pylori* infection is most prevalent in Africa (79.1%), followed by Latin America (63.4%) and Asia (54.7%) [4]. In Indonesia, *H. pylori* infection prevalence is about 22.1%, suggesting *H. pylori* infects approximately one out of five of the population [5]. *H. pylori* infection is significantly correlated with incidences of gastritis, gastroesophageal reflux disease, gastroduodenal ulcers, gastric mucosal-associated lymphoid tissue (MALT) lymphoma, and gastric malignancies [6,7,8,9]. The eradication of *H. pylori* is vital in reducing peptic ulcer recurrence, in the principal therapy of gastric MALT lymphoma, and in minimizing the risk of gastric cancer [10,11,12].

*H. pylori* elimination therapy commonly uses proton pump inhibitor (PPI)-based combination therapy for 7–14 days by combining a PPI and a minimum of two antibiotics, sometimes with the addition of bismuth. PPI takes a crucial role in *H. pylori* eradication by suppressing gastric acid secretion, hence enhancing antibiotics efficacies [13]. However, the success rate of PPI-based eradication therapy declines with antibiotics resistance emergence and inadequate acid suppression [13,14]. Increasing PPI dosage does not increase the eradication rate of PPI-based regimens [15,16,17]. Vonoprazan and tegoprazan are new potential gastric acid suppression agents, classified as potassium-competitive acid blockers (P-CAB), that function by H^+^/K^+^-ATPase inhibition [18,19] (Figure 1). Japanese guidelines on the management of *H. pylori* infections recommend replacing PPI with vonoprazan in first-line and second-line *H. pylori* eradication therapies since first introduced in 2015, while tegoprazan has been established as treatment for gastroesophageal reflux disease (GERD) in South Korea since 2018 [20,21]. Tegoprazan showed clinical benefits in phase-III studies for erosive esophagitis patients [22] and improved both gastric-related diseases and motility defects in a canine study [23]. However, studies of tegoprazan for *H. pylori* eradication are still in progress. Several non-randomized control trials (RCT), RCT, and meta-analyses reported encouraging results using vonoprazan-based therapies in eradicating *H. pylori*. Vonoprazan is expected to be a new candidate in *H. pylori* eradication regimens.

We collected all relevant studies after searching comprehensively using predefined keywords through the online databases of PubMed, Web of Science, EMBASE, and The Cochrane Library. We searched all relevant articles for vonoprazan-based eradication regimens (keywords: “vonoprazan” OR “VPZ” OR “TAK-438” OR “Potassium-Competitive Acid Inhibitor” AND “*Helicobacter pylori*” OR “*H. pylori*”) and for PPI-based eradication regimens (keywords: “proton pump inhibitor” OR “PPI” OR “omeprazole” OR “lansoprazole” OR “esomeprazole” OR “rabeprazole” AND “*Helicobacter pylori*” OR “*H. pylori*”). We first screened the title and abstract of each study and then examined full text against our inclusion criteria. We included all articles about comparative, retrospective, RCT, and meta-analysis studies of *H. pylori* eradication therapies in human populations using both regimens until April 2020. We extracted data about vonoprazan-based and PPI-based regimens with their dosages and *H. pylori* eradication rates. Our exclusion criteria are animal and Non-English studies.

Unsatisfactory acid-suppressing therapy outcomes prior to PPI discovery and development expedited research in obtaining new therapeutic agents. Initial studies revealed PPI has a higher efficacy than histamine-2 receptor antagonist-based therapies [24]. Table 1 reviews *H. pylori* eradication regimens approved by several gastroenterological societies.

Unfortunately, the clinical effectiveness of PPI-based regimens are reduced by antibiotic resistance. Failure of first-line eradication therapy is caused by the emergence of clarithromycin-resistant *H. pylori* with a failure rate of 60–70% [28,29]. Otherwise, metronidazole-resistant *H. pylori* is the main cause of second-line eradication therapy, especially in Southeast Asia [30]. Resistance to levofloxacin has emerged in some countries at a resistance rate of 20–40% [31,32,33]. As established previously, increasing PPI doses does not improve the eradication rate significantly. Consequently, vonoprazan was introduced as a PPI substitution candidate in all *H. pylori* eradication regimens as given by the Japanese guidelines [20].

## 2. Clinical Benefits of Vonoprazan

### 2.1. Pharmacological Aspects

Vonoprazan is acid-stable and can act as fast-release therapy, with a maximum plasma concentration (C_max_) that increases from 10 to 60 ng/mL in only 1.5–2 h [34,35]. Furthermore, it has an area under curve (AUC) from no time to infinity in a dose range of 1.14–1.32 ng.h/mL and is significantly influenced by intestinal meal absorption [34,35,36]. Although there are no significant differences in holding time ratio at pH > 4 and time elapsed to reach C_max_, vonoprazan has higher salutary C_max_, AUC, and half-life compared with PPI. Its negative logarithm of acid dissociation constant (pKa) > 9.0 as it is more concentrated in the secretory canaliculi of gastric parietal cells than in plasma [36,37] and has higher positive charged points [38]. Its distribution depends on albumin and alpha-1 acid glycoprotein [34].

Unlike PPI, vonoprazan does not require acid activation. It is primarily metabolized in the liver by cytochrome P450 3A4 (CYP 3A4) and metabolized partially by CYP2B6, CYP2C19, CYP2D6, and SULT2A1 [21,39]. Pharmacokinetic interaction between vonoprazan and clarithromycin is synergic because clarithromycin is a strong CYP3A4 inhibitor, which thus reduces vonoprazan metabolism [40]. Conversely, PPI is metabolized primarily by CYP2C19, which has an extensive metabolizer polymorphism that affects PPI efficacies and the prodrug activation process [21,41]. Research on acid suppression agents developed dramatically after the discovery of the crucial role of H^+^/K^+^-ATPase in the last stage of gastric acid secretion. PPI is a prodrug activated by acid and forms disulfide bonds with the cysteine component of H^+^/K^+^-ATPase [37,42]. PPI reaches maximum acid stability after 3–5 days of treatment [43,44].

The inability of PPI to create a gastric base environment has led researchers to investigate alternative acid-suppressing agents. One possible alternative mechanism is the reduction of potassium ion concentration to limit H^+^/K^+^-ATPase efficacy. P-CAB agents, such as vonoprazan, act as a reversible competitive inhibitors against potassium ions by binding with H^+^/K^+^-ATPase [45,46]. Vonoprazan is stable in the acidic gastric secretory canaliculi environment and binds non-covalently to H^+^/K^+^-ATPase [47]. Vonoprazan dissociates gradually and represses H^+^/K^+^-ATPase formation for a sustained period, consequently increasing gastric pH up to pH 7 in approximately in 4 h [48]. The differences between the pharmacokinetics and pharmacodynamics of PPI and Vonoprazan are shown in Table 2 [44,46].

Vonoprazan has potential to substitute PPI in GERD and gastroduodenal ulcer management. Standard therapy for these diseases is PPI, yet the outcomes are unsatisfactory. Substituting PPI for vonoprazan in erosive esophagitis can relieve symptoms quickly and significantly. A meta-analysis proved the superiority of vonoprazan against PPI in GERD management and subgroup analysis noted that vonoprazan significantly has higher efficacy in treating erosive esophagitis [49], especially in CYP2C19 EM patients with an efficacy rate of 90.0%, compared with 79.3% for PPI [50]. Vonoprazan is also superior against lansoprazole in treating peptic ulcer, with recurrence rates at 3.3% vs. 5.5%, respectively, as confirmed by endoscopy examination [51]. An RCT study confirmed the high efficacy of vonoprazan in peptic ulcer treatment, at 93.5% compared with lansoprazole at 93.8%. However, the study could not establish its efficacy in duodenal ulcer treatment due to dropped out patients and recurring ulcers [52]. Vonoprazan also has comparable efficacy with lansoprazole in reducing peptic ulcer recurrence incidence in patients taking low-dose aspirin [53]. Furthermore, a meta-analysis showed that patients who received vonoprazan for peptic ulcer related to endoscopic gastric submucosal resection have statistically significantly higher healing rates compared with those who received PPI (pooled odds ratio (OR) 2.27, 95% CI 1.38–3.73, heterogeneity (I^2^) = 0%, *p* = 0.001) [54].

### 2.2. Vonoprazan and H. pylori Eradication

*H. pylori* eradication is essential for preventing and intervening in long-term complications. Determinants influencing the eradication rate of *H. pylori* eradication therapy include antibiotic resistance, acid suppression adequacy, virulence factors (*cagA*, *vacA*, *dupA*), and the environment [55,56,57,58]. PPI-based *H. pylori* eradication therapy regimens have been ineffective, and the efficacy of doubling the PPI dose has little evidence and weak recommendations [17]. In addition, the CYP2C19 extensive metabolizer polymorphism diminishes PPI ability in suppressing gastric acid. 

Vonoprazan is a strong candidate replacement for PPI in *H. pylori* eradication regimens. Its pharmacological advantages include no requirement of acid activation, stability in an acidic environment, and a longer half-life [55]. Standardized first-line *H. pylori* eradication therapy is PPI, clarithromycin, and amoxicillin. RCT and non-RCT studies have revealed that vonoprazan-based eradication regimens have a higher eradication rate than PPI-based regimens (Table 3). Our previous meta-analysis of five clarithromycin-sensitive *H. pylori* RCT studies revealed no statistically significant differences between the eradication rates of first-generation PPI-based (pooled risk ratio (RR) 1.01, 95% CI 0.98–1.04, I^2^ = 61%, *p* = 0.04) and vonoprazan-based regimens (pooled RR 0.84, 95% CI 0.57–1.25, I^2^ = 0%, *p* = 0.39). However, we found significant differences between the eradication rates of second-generation PPI-based (pooled RR 1.25, 95% CI 1.15–1.37, I^2^ = 82%, *p* < 0.00001) and vonoprazan-based regimens (pooled RR 0.31, 95% CI 0.23–0.42, I^2^ = 50%, *p* < 0.00001), as well as between all PPI generation-based (pooled RR 1.11, 95% CI 1.07–1.16, I^2^ = 98%, *p* < 0.00001) and vonoprazan-based regimens combined (pooled RR 0.43, 95% CI 0.34–0.55, I^2^ = 81%, *p* < 0.00001) [15]. Several studies on vonoprazan-based regimens showed higher eradication rates against clarithromycin-resistant *H. pylori* (Table 4). Furthermore, a meta-analysis concluded that vonoprazan-based regimens have superiority in eradicating clarithromycin-resistant *H. pylori* (pooled eradication rates 82% and 40%, pooled OR 6.83, 95% CI 3.63–12.86, I^2^ = 0%, *p* < 0.0001) [59].

Standard second-line *H. pylori* eradication therapy, which is used after the failure of first-line therapy, consists of PPI, amoxicillin, and metronidazole. We did not find any RCT studies comparing the outcomes of vonoprazan-based and PPI-based second-line *H. pylori* eradication therapies (Table 5). Shinozaki et al. conducted a meta-analysis of non-RCT studies and concluded that vonoprazan-based second-line eradication regimens are statistically significant in eradicating *H. pylori* (pooled OR 1.51, 95% CI 1.27–1.81, I^2^ = 0%, *p* < 0.00001) [76].

Third-line *H. pylori* eradication regimens combine PPI or vonoprazan with amoxicillin and sitafloxacin. A study revealed that the third-line vonoprazan-based regimen has a higher *H. pylori* eradication rate than the PPI-based regimen (75.8% vs. 53.3%) [78]. Another study also showed the vonoprazan-based regimen has a higher eradication rate against sitafloxacin-resistant *H. pylori* than esomeprazole-based regimens (91.7% vs. 71.2%) [79]. Studies on third-line *H. pylori* eradication therapies are limited because third-line therapies are not covered under the universal Japanese health insurance [80].

The main limitation in this review is that all studies were conducted in Japan, which thus raises the question on the efficacy rates of vonoprazan outside Japan. For one, the Japanese population tends to have a higher holding time ratio at pH > 4 than the UK population [35,36]. For another, every region has a different antibiotic resistance mapping. For example, Japan has a high clarithromycin resistance rate (>30%) but a low metronidazole resistance rate (<5%) [81]. A contradictory study conducted in Indonesia revealed that *H. pylori* in the country has low clarithromycin resistance (9.1%) but high metronidazole and levofloxacin resistances at the rates of 46.7% and 31.2%, respectively [33].

A high incidence of *H. pylori* with poly-antimicrobial resistances compels research into alternative *H. pylori* eradication therapies. Previously, we investigated alternative therapies against metronidazole-resistant and levofloxacin-resistant *H. pylori* strains in Indonesia, Bangladesh, and Bhutan in vitro and found that furazolidones, rifaximin, rifabutin, garenoxacin, and sitafloxacin are effective in eradicating *H. pylori* [33,82]. Alternative therapy using herbal medicine such as the Indian plant *Bombax ceiba*, or a propolis (*Trigona* sp.) ethanol extract can inhibit the growth of metronidazole-resistant and levofloxacin-resistant *H. pylori* in vitro [83,84].

### 2.3. Safety and Adverse Events

Since the discovery of P-CAB, the most recognized complication is hepatotoxicity, although no serious adverse effects have been observed [42,44]. Unlike the previous P-CAB group, which is a derivative of an imidazole–pyridine compound, vonoprazan is a pyridine-derivative compound, which has a lower hepatotoxicity risk [21,85]. Nevertheless, some previous studies did not encounter any significant difference between transaminase increases in patients receiving vonoprazan and PPI [34].

The acid inhibition rate of vonoprazan is better than that of PPI. Consequently, the increase in gastrin serum in patients receiving vonoprazan therapy is higher than in patients receiving PPI therapy [16,50]. Hypergastrinemia can trigger gastric enterochromaffin cell hyperplasia and increase the risk of gastric endocrine tumors [86,87]. Hypochlorhydria precipitated by acid inhibition can alter the gut microbiome, increasing the risk of developing antibiotic-associated diarrhea caused by *Clostridium difficile* and spontaneous bacterial peritonitis [88,89]. Excessive acid suppression can also cause malabsorption, resulting in the onset of iron deficiency anemia, megaloblastic anemia, hypomagnesia, and hypocalcemia [46,90]. Additional side effects that can emerge include interstitial nephritis, pneumonia, dementia, chronic kidney disease, and ischemic heart disease [91,92,93].

## 3. Conclusions

Vonoprazan is a promising alternative to PPI in the treatment of gastroduodenal diseases, mainly in *H. pylori* eradication therapy, given its superior pharmacological and clinical indications to PPI. However, further clinical studies on vonoprazan are required to confirm its efficacies, especially outside Japan, in order to establish universally conclusive results.

## Figures and Tables

**Figure 1 pharmaceuticals-13-00276-f001:**
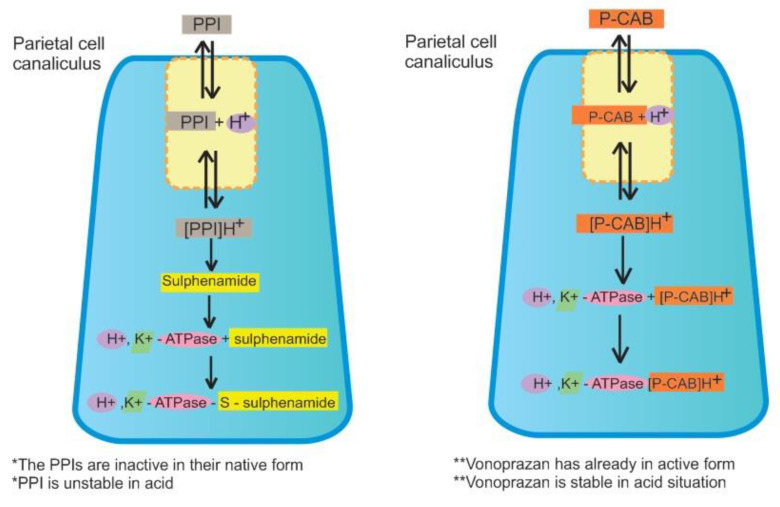
The mode of action of proton pump inhibitors (PPIs) and Vonoprazan against *H. pylori* infection. PPI enters parietal cell canaliculus in inactive form and requires acid activation. Protonated pro-drugs will convert to sulphenamide and bond covalently with cysteine groups of H^+^/K^+^-ATPase that cause inactivation of H^+^/K^+^-ATPase. Unlike PPI, potassium-competitive acid blocker (P-CAB) enters parietal cell canaliculus in active form, has stability in an acidic environment, and does not require acid activation. Protonated P-CAB will make non-covalent bonds with H^+^/K^+^-ATPase, thus inactivating the H^+^/K^+^-ATPase with a slower dissociation rate and for longer time. * Lacks of PPIs in gastric acid inhibition; ** Benefits of vonoprazan in gastric acid inhibition.

**Table 1 pharmaceuticals-13-00276-t001:** *Helicobacter pylori* eradication therapy regimens based on several guidelines.

	American College of Gastroenterology [25] Indonesian Society of Gastroenterology [26]	Japanese Society for *Helicobacter* Research [20]	The Toronto Consensus [27]	The Maastricht V/Florence Consensus Report [17]
First Line	If Clarithromycin-resistant strains < 20% PPI * Amoxicillin Clarithromycin If Clarithromycin-resistant strains > 20% PPI * Bismuth subsalicylate Metronidazole Tetracycline	PPI * or Vonoprazan Amoxicillin Clarithromycin	If Clarithromycin-resistant strains < 15% PPI * Amoxicillin/Metronidazole Clarithromycin If Clarithromycin-resistant strains > 15% PPI * Amoxicillin Metronidazole Clarithromycin or PPI * Bismuth Metronidazole Tetracycline	If Clarithromycin-resistant strains < 15% PPI * Amoxicillin Clarithromycin If Clarithromycin-resistant strains > 15% PPI * Amoxicillin Metronidazole
Second Line	PPI * Bismuth subsalicylate Metronidazole or PPI * Amoxicillin Levofloxacin	PPI * Amoxicillin Metronidazole	PPI * Amoxicillin Levofloxacin or PPI * Bismuth Metronidazole Tetracycline	PPI * Amoxicillin Levofloxacin or PPI * Bismuth Metronidazole Tetracycline
Third Line	PPI * Amoxicillin Levofloxacin Rifabutin	PPI * Amoxicillin/Metronidazole Sitafloxacin	PPI * Amoxicillin Rifabutin	Regimens based on the bacterial culture susceptibility test

* PPI agents used are Omeprazole 20 mg, Lansoprazole 30 mg, Esomeprazole 40 mg, Rabeprazole 20 mg, Pantoprazole 40 mg.

**Table 2 pharmaceuticals-13-00276-t002:** Pharmacological comparisons between PPI and Vonoprazan.

Parameter	PPI		Vonoprazan
	First Generation PPI	Second Generation PPI	
Acid activation	Yes		No
Active drug	No		Yes
Acid Stability	No		Yes
Main P450 metabolizer	CYP2C19		CYP3A4
Meal’s influence	Yes		No
Mechanism of Action	Covalent bond to gastric proton pump		Potassium ion competitive reversible inhibitor to gastric proton pump
Days required for reaching maximal acid suppression	3–5		1
pH > 4 holding time (%)	OMZ 30.4 LPZ 39.1	ESO 43.1 RPZ 42.8	10 mg 38.4–43.1 20 mg 62.7–63.3
Time Needed to Reach Maximum Plasma Concentration (h)	OMZ 1–4 LPZ 1.2–2.1	ESO 1–3.5 RPZ 1.14	10 mg 1.75 20 mg 1.50
Half-life (h)	OMZ 0.5–1.2 LPZ 0.9–2.1	ESO 1.3–1.6 RPZ 0.6–1.4	10 mg 6.95 ± 1.03 20 mg 6.85 ± 0.80
C_max_ (μmol/L)	OMZ 0.23–23.2 LPZ 1.62–3.25	ESO 2.1–2.4 RPZ 1.14	10 mg 9.7 ± 2.1 μg/L 20 mg 25.0 ± 5.6 μg/L
AUC (μmol.h/L)	OMZ 0.58–3.47 LPZ 4.60–13.5	ESO 4.2 RPZ 2.22	10 mg 60.1 ± 9.0 μg.h/L 20 mg 160.3 ± 38.6 μg.h/L

AUC: Area Under Curve; C_max_: Maximum Plasma Concentration; CYP: Cytochrome P450 OMZ: Omeprazole 20 mg; LPZ: Lansoprazole 30 mg; ESO: Esomeprazole 40 mg; RPZ: Rabeprazole 20 mg.

**Table 3 pharmaceuticals-13-00276-t003:** Review of Comparative Studies First-line *H. pylori* Eradication Therapy.

Study	VPZ-Based Regimen		PPI-Based Regimen	
	Regimen	Eradication Rate	Regimen	Eradication Rate
RCT				
Murakami et al., 2016 [60]	VPZ: 20 mg bid AMX: 750 mg bid CLR: 200 or 400 mg bid	90.9%	LPZ: 30 mg bid AMX: 750 mg bid CLR: 200 or 400 mg bid	75.1%
Maruyama et al., 2017 [61]	VPZ: 20 mg bid AMX: 750 mg bid CLR: 200 or 400 mg bid	95.8%	LPZ: 30 mg bid or RPZ: 20 mg bid AMX: 750 mg bid CLR: 200 or 400 mg bid	69.6%
Sue et al., 2017 [62]	VPZ: 20 mg bid AMX: 750 mg bid CLR: 200 or 400 mg bid	87.3%	LPZ: 30 mg bid, RPZ: 10 mg bid or ESO: 20 mg bid AMX: 750 mg bid CLR: 200 or 400 mg bid	76.5%
Ozaki et al., 2018 [63]	VPZ: 20 mg bid AMX: 750 mg bid CLR: 200 or 400 mg bid	90.9%	RPZ: 10 mg bid or ESO: 20 mg bid AMX: 750 mg bid CLR: 200 or 400 mg bid	72.8%
Non-RCT				
Suzuki et al., 2016 [64]	VPZ: 20 mg bid AMX: 750 mg bid CLR: 200 or 400 mg bid	89.0%	LPZ: 30 mg bid or RPZ: 20 mg bid AMX: 750 mg bid CLR: 200 mg bid	74.2%
Shinozaki et al., 2016 [65]	VPZ: 20 mg bid AMX: 750 mg bid CLR: 200 or 400 mg bid	82.9%	LPZ: 30 mg bid, RPZ: 10 mg bid or ESO: 20 mg bid AMX: 750 mg bid CLR: 200 mg bid	73.9%
Shichijo et al., 2016 [66]	VPZ: 20 mg bid AMX: 750 mg bid CLR: 200 or 400 mg bid	87.2%	LPZ: 30 mg bid, RPZ: 10 mg bid or ESO: 20 mg bid AMX: 750 mg bid CLR: 200 or 400 mg bid	72.4%
Noda et al., 2016 [67]	VPZ: 20 mg bid AMX: 750 mg bid CLR: 400 mg bid	89.7%	OMZ: 20 mg bid, LPZ: 30 mg bid, RPZ: 10 mg bid or ESO: 20 mg bid AMX: 750 mg bid CLR: 200 or 400 mg bid	73.9%
Matsumoto et al., 2016 [68]	VPZ: 20 mg bid AMX: 750 mg bid CLR: 200 mg bid	89.6%	LPZ: 30 mg bid, RPZ: 10 mg bid or ESO: 20 mg bid AMX: 750 mg bid CLR: 200 or 400 mg bid	71.9%
Yamada et al., 2016 [69]	VPZ: 20 mg bid AMX: 750 mg bid CLR: 200 mg bid	85.7%	LPZ: 30 mg bid, RPZ: 10 mg bid or ESO: 20 mg bid AMX: 750 mg bid CLR: 200 mg bid	73.2%
Tsujimae et al., 2016 [70]	VPZ: 20 mg bid AMX: 750 mg bid CLR: 200 mg bid	84.6%	ESO: 20 mg bid AMX: 750 mg bid CLR: 200 mg bid	79.1%
Kajihara et al., 2016 [71]	VPZ: 20 mg bid AMX: 750 mg bid CLR: 400 mg bid	94.6%	RPZ: 10 mg bid AMX: 750 mg bid CLR: 200 or 400 mg bid	86.7%
Sakurai et al., 2017 [72]	VPZ: 20 mg bid AMX: 750 mg bid CLR: 200 mg bid	87.9%	LPZ: 30 mg bid, RPZ: 10 mg bid or ESO: 20 mg bid AMX: 750 mg bid CLR: 200 mg bid	66.9%
Sue et al., 2017 [73]	VPZ: 20 mg bid AMX: 750 mg bid CLR: 200 or 400 mg bid	84.9%	OMZ: 20 mg bid, LPZ: 30 mg bid, RPZ: 10 mg bid or ESO: 20 mg bid AMX: 750 mg bid CLR: 200 or 400 mg bid	78.8%
Nishizawa et al., 2017 [74]	VPZ: 20 mg bid AMX: 750 mg bid CLR: 200 or 400 mg bid	62.3%	LPZ: 30 mg bid or RPZ: 10 mg bid AMX: 750 mg bid CLR: 200 or 400 mg bid	47.1%
Tanabe et al., 2018 [75]	VPZ: 20 mg bid AMX: 750 mg bid CLR: 200 or 400 mg bid	91.5%	LPZ: 30 mg bid, RPZ: 10 mg bid or ESO: 20 mg bid AMX: 750 mg bid CLR: 200 mg bid	79.4%

AMX: Amoxicillin, CLR: Clarithromycin, ESO: Esomeprazole, LPZ: Lansoprazole, OMZ: Omeprazole, RPZ: Rabeprazole, VPZ: Vonoprazan.

**Table 4 pharmaceuticals-13-00276-t004:** Review of comparative studies of first-line clarithromycin-resistant *H. pylori* eradication therapies.

Study	VPZ-Based Regimen		PPI-Based Regimen	
	Regimen	Eradication Rate	Regimen	Eradication Rate
RCT				
Murakami et al., 2016 [60]	VPZ: 20 mg bid AMX: 750 mg bid CLR: 200 or 400 mg bid	82.0%	LPZ: 30 mg bid AMX: 750 mg bid CLR: 200 or 400 mg bid	40.0%
Non-RCT				
Noda et al., 2016 [67]	VPZ: 20 mg bid AMX: 750 mg bid CLR: 400 mg bid	87.5%	OMZ: 20 mg bid, LPZ: 30 mg bid, RPZ: 10 mg bid or ESO: 20 mg bid AMX: 750 mg bid CLR: 200 or 400 mg bid	53.8%
Matsumoto et al., 2016 [68]	VPZ: 20 mg bid AMX: 750 mg bid CLR: 200 mg bid	76.1%	LPZ: 30 mg bid, RPZ: 10 mg bid or ESO: 20 mg bid AMX: 750 mg bid CLR: 200 or 400 mg bid	40.2%

AMX: Amoxicillin, CLR: Clarithromycin, ESO: Esomeprazole, LPZ: Lansoprazole, OMZ: Omeprazole, RPZ: Rabeprazole, VPZ: Vonoprazan.

**Table 5 pharmaceuticals-13-00276-t005:** Review of comparative studies of second-line *H. pylori* eradication therapies.

Study	VPZ-Based Regimen		PPI-Based Regimen	
	Regimen	Eradication Rate	Regimen	Eradication Rate
Yamada et al., 2016 [69]	VPZ: 20 mg bid AMX: 750 mg bid MNZ: 250 mg bid	89.6%	LPZ: 30 mg bid, RPZ: 10 mg bid or ESO: 20 mg bid AMX: 750 mg bid MNZ: 250 mg bid	89.9%
Tsujimae et al., 2016 [70]	VPZ: 20 mg bid AMX: 750 mg bid MNZ: 250 mg bid	89.1%	ESO: 20 mg bid AMX: 750 mg bid MNZ: 250 mg bid	83.3%
Sakurai et al., 2017 [72]	VPZ: 20 mg bid AMX: 750 mg bid MNZ: 250 mg bid	96.1%	LPZ: 30 mg bid, RPZ: 10 mg bid or ESO: 20 mg bid AMX: 750 mg bid MNZ: 250 mg bid	89.7%
Sue et al., 2017 [73]	VPZ: 20 mg bid AMX: 750 mg bid MNZ: 250 mg bid	80.5%	LPZ: 30 mg bid, RPZ: 10 mg bid or ESO: 20 mg bid AMX: 750 mg bid MNZ: 250 mg bid	81.5%
Nishizawa et al., 2017 [77]	VPZ: 20 mg bid AMX: 750 mg bid MNZ: 250 mg bid	71.8%	LPZ: 30 mg bid or RPZ: 10 mg bid AMX: 750 mg bid MNZ: 250 mg bid	73.7%

AMX: Amoxicillin, CLR: Clarithromycin, ESO: Esomeprazole, LPZ: Lansoprazole, MNZ: Metronidazole, RPZ: Rabeprazole, VPZ: Vonoprazan.

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
