# Peer review of "The Potential Benefits of Vonoprazan as Helicobacter pylori Infection Therapy"

_pharmaceuticals, 2020, doi:10.3390/ph13100276_

Round 1

Reviewer 1 Report

The review by Miftahussurur et al. captures an interesting aspect of using Vonoprazan as potentially better alternative for proton pump inhibitor in anti-Helicobacter pylori. Unfortunately, the style and language used in the review make it difficult to follow. I would also recommend to add a figure that will depict the mode of action of Vonoprazan against H. pylori infection. The table 2 is also not very readible.  

Author Response

We appreciate the thoughtful comments from the reviewers. The manuscript was fully evaluated in consideration of the reviewer’s individual comments. We revised Table 2, added Figure 1 and removed study by Takimoto et al. We also added information about The Toronto Consensus, the Maastricht V/Florence Consensus Report and Tegoprazan. We believe that their suggested changes helped us to substantially improve our manuscript.

Reviewer: The review by Miftahussurur et al. captures an interesting aspect of using Vonoprazan as potentially better alternative for proton pump inhibitor in anti-Helicobacter Unfortunately, the style and language used in the review make it difficult to follow. I would also recommend to add a figure that will depict the mode of action of Vonoprazan against H. pylori infection. The table 2 is also not very readible. 

Response to reviewer: Thank you for your comments. We have improved our English in the entire of the manuscript by native English (Enago™ Proofreading Services, New York, US) and fixed the table 2 to make it more readable (Page 4 line 159). It would be more attractive to have a figure in our manuscript, thus we have added a figure how Vonoprazan’s role differ from PPI in H. pylori infection eradication (Figure 1).

Reviewer 2 Report

This paper is a review of the H.pyori eradication using Vonoprazan. However, although there were some descriptions of keywords and exclusion criteria, there was no description of the process by which the final evaluated paper was selected. Did the authors consider the content of each paper accurately? In addition, the eradication rate of the report of Takimoto et al. cited in Table 3 is extremely low, but the literature has not examined the eradication rate.  This may lead to doubts about the reliability of the entire paper.

Author Response

We appreciate the thoughtful comments from the reviewers. The manuscript was fully evaluated in consideration of the reviewer’s individual comments. We revised Table 2, added Figure 1 and removed study by Takimoto et al. We also added information about The Toronto Consensus, the Maastricht V/Florence Consensus Report and Tegoprazan. We believe that their suggested changes helped us to substantially improve our manuscript.

Reviewer: This paper is a review of the H.pyori eradication using Vonoprazan. However, although there were some descriptions of keywords and exclusion criteria, there was no description of the process by which the final evaluated paper was selected. Did the authors consider the content of each paper accurately? In addition, the eradication rate of the report of Takimoto et al. cited in Table 3 is extremely low, but the literature has not examined the eradication rate. This may lead to doubts about the reliability of the entire paper.

Response to reviewer: Thank you for your comments. We made our methods in arranging this manuscript more detailed, especially what data we extracted (Page 2 lines 69-73). Thank you for your correction, we removed study by Takimoto et al from our inclusion study as the primary outcome in the study was change of UBT level, while the primary outcome extracted in our manuscript is eradication rate

Reviewer 3 Report

All my comments have been addressed. I have no further comments.

Author Response

All my comments have been addressed. I have no further comments.

Response to reviewer: Thank you very much for your comments. We have improved our English in this manuscript (Enago™ Proofreading Services, New York, US).

Reviewer 4 Report

  1. The guideline referenced when creating Table 1 is insufficient. Please modify the table by adding more references (The Toronto Consensus for the Treatment of Helicobacter pylori Infection in Adults, Management of Helicobacter pylori infection-the Maastricht V/Florence Consensus Report).
  2. Paragraphs 2.2 and 2.3 are not required in this review article (vonoparzan and GERD or peptic ulcer). Please shorten the above paragraphs into one paragraph.
  3. Please also add a description of tegoprazan, a novel P-CAB similar to Vonoprazan.

Author Response

  1. The guideline referenced when creating Table 1 is insufficient. Please modify the table by adding more references (The Toronto Consensus for the Treatment of Helicobacter pylori Infection in Adults, Management of Helicobacter pylori infection-the Maastricht V/Florence Consensus Report).
  1. Paragraphs 2.2 and 2.3 are not required in this review article (vonoparzan and GERD or peptic ulcer). Please shorten the above paragraphs into one paragraph.
  2. Please also add a description of tegoprazan, a novel P-CAB similar to Vonoprazan.

Response to reviewer: Thank you for your comments. We have added H. pylori eradication regimens based on Japanese Society for Helicobacter Research, The Toronto Consensus for the Treatment of Helicobacter pylori Infection in Adults, and Management of Helicobacter pylori infection-the Maastricht V/Florence Consensus Report (Pages 2-3 lines 80). We have summarized the paragraphs about the use of Vonoprazan for GERD and peptic ulcers in to one paragraph. We also added a description of Tegoprazan as one of P-CAB agent (Page 2 lines 60).

Round 2

Reviewer 1 Report

The manuscript of Miftahussurur et al. has been significally improved comparing to the previous version. Nevertheless, better description of the Figure 1 is needed, because it is unclear what exactly is displayed. The description can be placed in the legend of the picture.

Author Response

The manuscript of Miftahussurur et al. has been significally improved comparing to the previous version. Nevertheless, better description of the Figure 1 is needed, because it is unclear what exactly is displayed. The description can be placed in the legend of the picture.

Response to reviewer: Thank you very much for your comments and suggestions in giving much improvements for this paper. We have added the description for the Figure 1 and hopefully can give better explanation for the figure.

Reviewer 2 Report

The revised paper is appropriate as a review of “The Potential Benefits of Vonoprazan as Helicobacter pylori Infection Therapy”.

Author Response

  1. The revised paper is appropriate as a review of “The Potential Benefits of Vonoprazan as Helicobacter pylori Infection Therapy”.

Response to reviewer: Thank you very much for your comments and suggestions in giving much improvements for this paper.